# QTL Mapping of Yield-Related Traits in Tetraploid Wheat Based on Wheat55K SNP Array

**DOI:** 10.3390/plants13101285

**Published:** 2024-05-07

**Authors:** Yatao Jia, Yifan Zhang, Yingkai Sun, Chao Ma, Yixiong Bai, Hanbing Zhang, Junbin Hou, Yong Wang, Wanquan Ji, Haibo Bai, Shuiyuan Hao, Zhonghua Wang

**Affiliations:** 1State Key Laboratory for Crop Stress Resistance and High-Efficiency Production, College of Agronomy, Northwest A&F University, Yangling 712100, China; sxndjyt@163.com (Y.J.); 13379314638@163.com (Y.Z.); sykdoican@163.com (Y.S.); 13271735195@163.com (C.M.); 18829353496@163.com (H.Z.); houjunbin123@163.com (J.H.); wangyong2114@163.com (Y.W.); jiwanquan2003@126.com (W.J.); 2Qinghai Key Laboratory of Hulless Barley Genetics and Breeding, Qinghai Academy of Agricultural and Forestry Sciences, Qinghai University, Xining 810016, China; yixiongbai@163.com; 3Agricultural Bio-Technology Research Center, Ningxia Academy of Agriculture and Forestry Science, Yinchuan 750002, China; bai-haibo@163.com; 4Department of Agriculture, Hetao College, Bayan Nur City 015000, China

**Keywords:** 55K SNP array, tetraploid wheat, yield-related traits, QTL

## Abstract

To enhance the understanding of yield-related traits in tetraploid wheat, it is crucial to investigate and identify genes that govern superior yield characteristics. This study utilized the wheat55K single nucleotide polymorphism array to genotype a recombinant inbred line (RIL) population consisting of 120 lines developed through the crossbreeding of two tetraploid wheat varieties, Qin Hei-1 (QH-1) and Durum Wheat (DW). An investigation and analysis were conducted on 11 yield-related traits, including peduncle length (PL), neck length (NL), spike length (SL), flowering date (FD), heading date (HD), thousand-kernel weight (TKW), kernel area ratio (KAR), kernel circumference (KC), kernel length (KL), kernel width (KW), and kernel length–width ratio (KL-WR), over a period of three years in two locations, Yang Ling, Shaanxi, and Lin He, Inner Mongolia. The analysis identified nine stable loci among eight agronomic traits, named *QSL.QD-1A.1*, *QNL.QD-4B.2*, *QPL.QD-4B.1*, *QFD.QD-2B*, *QHD.QD-2B.1*, *QHD.QD-4B*, *QKC.QD-4B.2*, *QKL-WR.QD-4B.6*, and *QKL.QD-4B.2*. Among them, the additive effects of three QTLs, *QSL.QD-1A.1*, *QNL.QD-4B.2*, and *QFD.QD-2B*, were positive, indicating that the enhancing alleles at these loci were derived from the parent line QH-1. These three QTLs showed significant positive effects on the phenotypes of the population materials. Furthermore, potential functional genes were identified within the mapping intervals of *QSL.QD-1A.1* and *QNL.QD-4B.2*, which regulate the development of spike length and neck length, respectively. These results provide potential QTLs and candidate genes, which broaden the genetic basis of agronomic traits related to yield, such as SL, NL, PL, and FD, and benefits for wheat breeding and improvement.

## 1. Introduction

Wheat yield, a complex trait influenced by numerous genes and the interaction of environmental and genetic factors throughout plant growth stages [1], encompasses various components, including plant morphology such as peduncle length (PL), neck length (NL), and spike length (SL). Additionally, yield traits are influenced by growth stage indicators like flowering date (FD) and heading date (HD), along with grain development-related characteristics such as thousand-kernel weight (TKW), kernel area ratio (KAR), kernel circumference (KC), kernel length (KL), kernel width (KW), and kernel length–width ratio (KL-WR). Peduncle length, a component of plant height representing the distance from the panicle base to the flag leaf petiole [2], plays a pivotal role in shaping overall plant architecture and is closely linked to inflorescence development and the transport of reserves to the spike and grain [3,4]. NL, a constituent of PL, denotes the distance from the internode at the base of the spike to the tip of the flag leaf sheath, specifically referring to the retracted portion of the flag leaf sheath. Through extensive breeding efforts, it has been observed that cultivars exhibiting a higher NL to PL ratio tend to display accelerated grain filling, enhanced senescence, earlier maturity, and increased grain weight per spike, suggesting a positive correlation between NL relative to PL and grain weight increment [5]. However, heightened plant stature resulting from increased PL and NL may predispose the plant to panicle bending or lodging, potentially leading to reduced grain yield and quality [6]. Furthermore, the metabolism and signaling of gibberellic acid (GA) significantly impact the development of PL and NL [3,7]. Therefore, PL and NL are often closely related to plant height genes. However, there are exceptions to this rule. At present, there are two site independent of plant height gene on 5AS and 6D for PL trait, providing a chance to modulate the PL [8,9]. Spike architecture traits such as spike length (SL) are closely related to grain production in wheat [10]. A number of studies have identified stable QTLs for SL on various chromosomes of hexaploid wheat [11]. It has been reported that the Q gene on chromosome 5A, which encodes an AP2 transcription factor, affects SL in wheat [12]. 

FD and HD not only govern the duration of wheat dry matter accumulation but also play a significant role in enhancing wheat’s adaptability to varying environmental conditions [13]. Over 100 quantitative trait loci (QTLs) associated with HD have been identified across all wheat chromosomes [11]. Studies have shown that vernalization demand and photoperiod sensitivity are the main factors influencing and controlling HD genetics [14]. *VRN1*, *VRN2*, *VRN3*, and *VRN4* are the main vernalization genes affecting wheat vernalization [15,16,17,18]. The three homologous dominant loci Ppd-A1, Ppd-B1, and Ppd-D1, which mainly respond to photoperiodic regulation, are situated on chromosomes 2A, 2B, and 2D, respectively [19]. 

Kernel traits encompass several characteristics such as KL, KW, KC, KAR, TKW, and KL-WR. KL is predominantly determined during the initial stages of kernel development, exhibiting lower susceptibility to environmental influences [20,21]. In contrast, KW and KC are established later in development and are more responsive to environmental conditions. Recent investigations have identified numerous quantitative trait loci (QTLs) associated with kernel traits distributed across all 21 chromosomes of wheat [11,22]. These kernel traits are closely interrelated and directly impact both yield and end-use quality of wheat [23]. Consequently, comprehending the genetic mechanisms governing wheat yield-related traits is of paramount importance for enhancing wheat varieties and developing novel cultivars with superior yield and quality attributes.

Common wheat (*Triticum aestivum*) is the most widely cultivated cereal crop globally and is a typical allohexaploid (AABBDD), formed by the hybridization and genome duplication of cultivated tetraploid wheat (AABB) and the diploid wild goatgrass (DD). With human migration activities, it has spread worldwide. Therefore, the genome of hexaploid common wheat can be traced back to tetraploid cultivated wheat, ultimately tracing back to the wild diploid wheat lineage [24]. Synthetic hexaploid wheat (SHW) is a valuable genetic resource that can enhance the performance of common wheat by transferring favorable genes from a wide range of tetraploid or diploid donors [25]. Previous studies have shown that hexaploid wheat contains six HMW-GS genes (Glu-1) located on chromosomes 1A, 1B, and 1D, whereas tetraploid wheat has four HMW-GS loci on chromosomes 1A and 1B. The main difference is the absence of HMW glutenin subunits from 1D in tetraploid wheat. However, three classes of HMW-GS protein exist in tetraploid wheat: 1Ax (encoded by *Glu-A1*), and 1Bx and 1By (encoded by *Glu-B1*), while hexaploid wheat has five classes: 1Ax (encoded by *Glu-A1*), 1Bx and 1By (encoded by *Glu-B1*), and 1Dx and 1Dy (encoded by *Glu-D1*), showing genetic gene sharing between tetraploid and hexaploid wheat [26]. Therefore, in the mapping of quantitative traits, especially yield-related traits, comparisons and tracing between the genomes of diploid, tetraploid, and hexaploid wheat are often conducted. For instance, Mo Z et al. predicted the involvement of some development-related genes in spikelet growth and their impact on spikelet number per spike by comparing the genetic positions of tetraploid Triticum turgidum and hexaploid Chinese Spring [27]. Liu J et al. identified twenty common genes in the physical interval between flanking markers on chromosome 4B of “Chinese Spring” and wild emmer, which regulate productive tiller number [28]. Considering the phenomenon of shared genetic variation in yield-related traits between tetraploid and hexaploid wheat, and the relative simplicity of the tetraploid wheat genome, research on the localization of yield-related traits in tetraploid wheat is beneficial for accelerating the breeding process of common wheat.

QH-1, characterized as a spring wheat variant, exhibits notable phenotypic traits and resistance attributes, including a compact plant stature of approximately 90 cm, increased tillering, slender grains, and robust disease resistance. In contrast, DW, a tetraploid wheat variety, holds the second position after common wheat and accounts for approximately 5% of the global annual wheat yield. DW is extensively cultivated across Europe, North Africa, and West Asia, where it is predominantly used in local culinary practices [29,30]. The plants are shorter in height, with sturdy stems that are resistant to lodging, and have larger grains. Furthermore, they exhibit strong stress resistance and high grain hardness, rendering them exceptional parental lines for genetic breeding. This study utilized a RIL population derived from two tetraploid wheat materials, “QH-1” and “DW”, as parents. The investigation focused on 11 yield-related traits over a period of three years in two environmental locations, Yang Ling, Shaanxi, and Lin He, Inner Mongolia. By combining the survey results of the parents and RIL population materials with genotyping using the wheat55K SNP chip strategy, additive QTL analysis was conducted in single and multiple environments, aiming to identify the major genes or loci influencing yield-related traits in tetraploid wheat. This study conducted gene mining and material characteristic evaluation on two levels to provide a theoretical basis for genetic improvement of yield-related traits in wheat.

## 2. Results

### 2.1. Phenotypic Differences and Analysis

To assess variances among materials, an examination of 11 traits within the recombinant inbred line (RIL) population comprising parental and hybrid combinations was conducted. The findings indicated a marked elevation in the lengths of PL, NL, and SL linked to yield in QH-1 in contrast to DW and KW in DW, which significantly surpassed that of QH-1, with no observable change in KL. Consequently, it is evident that TKW, KC, and KAR in DW exhibit slight superiority over those in QH-1 (Figure 1 and Table 1).

Through BLUE analysis of parental and RIL populations across five environmental time points (2021-2023YL and 2022-2023LH), followed by normal distribution assessment, it was observed that all 11 yield-associated traits displayed a continuous distribution and transgressive segregation, indicative of quantitative trait attributes governed by multiple genes (Table 1 and Appendix A). Examination of population phenotypes via *H*^2^ analysis unveiled that FD, HD, and KAR are prone to environmental influences, whereas the remaining traits are predominantly determined by genetic factors (Table 1).

### 2.2. Correlation Analyses among Different Traits

The phenotypic correlations among the 11 agronomic traits, as determined by BLUE values, are visually depicted in Figure 2. The analysis indicates notably strong positive correlations (*p* < 0.01) between PL and NL; HD and FD; TKW and SL, NL, PL; KC and SL, TKW; KL-WR and FD, HD, KC; KL and SL, TKW, KC, KL-WR; KW and SL, NL, PL, TKW, KC, KL. Notably, the correlation coefficient between KC and KL is the most pronounced at 0.98, followed by 0.94 between PL and NL. Furthermore, a significant negative correlation (*p* < 0.01) is observed between KL-WR and NL, PL, and KW, with the strongest correlation coefficient being −0.49. Collectively, the holistic evaluation of yield traits indicates a marked positive correlation among the majority of traits within the population, exerting a supportive role in the developmental trajectory.

### 2.3. Genetic Map Construction

Genotyping of all 120 families of RILs was performed using the Wheat 55K SNP chip. Following thorough filtration through software and manual curation, a total of 11,704 pure and polymorphic SNP markers differing between the parental lines were identified after excluding SNPs with unsuccessful genotype calls. Subsequently, employing the BIN function of IciMapping V4.1, measures were taken to eliminate missing values, segregational distortions surpassing the predefined threshold, and redundant markers, resulting in a refined collection of 913 high-quality SNP markers. The constructed genetic map spans a total length of 2141.26 cM, showcasing an average marker density of 2.35 cM per SNP. Through a comparative analysis of the probe information in the SUM file of the mapping outcomes with the genomic data of Chinese Spring wheat, a subset of 848 informative probes residing on the A/B chromosome groups was identified, distributed across 26 linkage groups spanning 14 chromosomes (Appendix A).

### 2.4. QTL Mapping Analysis

The ICIM-BIP approach was employed to identify additive QTLs associated with the 11 specified traits, leading to the detection of a total of 100 additive QTLs as presented in Appendix A. These QTLs exhibited a range of LOD values from 2.50 to 26.71 and were located on 14 chromosomes, explaining 0.23% to 55.42% of the phenotypic variance (Appendix A). Notably, within the identified additive QTLs, a subset of 41 were categorized as major QTLs, with 9 demonstrating stability across multiple environmental conditions, spanning eight distinct traits. These nine stable QTLs are named *QSL.QD-1A.1*, *QNL.QD-4B.2*, *QPL.QD-4B.1*, *QFD.QD-2B*, *QHD.QD-2B.1*, *QHD.QD-4B*, *QKC.QD-4B.2*, *QKL-WR.QD-4B.6*, and *QKL.QD-4B.2* (Table 2 and Appendix A, Figure 3).

### 2.5. Combined QTL–Environment Interaction Analysis

The MET-Add method was used for cQTL analysis, and a total of 259 additive cQTLs for SL (22), NL (26), PL (21), FD (10), HD (16), TKW (28), KAR (18), KC (27), KL-WR (42), KL (28), and KW (21) were mapped (Appendix A). Most of these cQTLs were minor QTLs. Moreover, all of the nine stable QTL detected using the ICIM-BIP method were similarly detected using the MET-Add method (Appendix A), showing that they might be stable in expression and less susceptible to environmental influences. Among the nine stable QTLs located on eight traits, the proportion of phenotypic variance explained (PVE) by the additive genetic effect (PVE (A)) is higher than PVE (A/E) for six QTLs associated with SL, PL, HD, KC, and KL traits. This indicates that the additive effects of these six stable major loci have a greater impact on the phenotypic variation compared to environmental effects, suggesting that genetic factors play a dominant role in phenotypic variation. Therefore, these loci can be utilized for genetic improvement to effectively enhance the quality traits. On the other hand, for NL, FD, and KL-WR traits, the PVE (A) is lower than PVE (A/E) for the stable major loci associated with these traits. This suggests that environmental effects have a greater influence on phenotypic variation compared to additive effects, indicating that environmental factors dominate the phenotypic variation for these traits.

In addition, the analysis results of the ICIM-BIP method and the MET-Add method both show that the additive effect values of *QSL.QD-1A.1*, *QNL.QD-4B.2*, *QPL.QD-4B.1*, and *QFD.QD-2B* in the nine stable QTLs are positive, indicating that the enhancing alleles at these four loci originate from the parent QH-1. The additive effect values of *QHD.QD-2B.1*, *QHD.QD-4B*, *QKC.QD-4B.2*, *QKL-WR.QD-4B.6*, and *QKL.QD-4B.2* are negative, indicating that the enhancing alleles at these five loci originate from the parent DW.

### 2.6. Analysis of the Effects of Stable Major Loci

To assess the impact of different genotypes on the associated phenotypes, families with predicted homozygous genotypes at relevant loci were identified using genotyping data generated by the wheat 55K SNP chip and genotypic values predicted by IciMapping. Families exhibiting plausible homozygous genotypes for the respective loci were selected. These families were then categorized into the Qin Hei (QH) type and Durum Wheat (DW) type based on the sources of homozygous alleles at the corresponding QTL loci within the population. Subsequently, the phenotypic traits linked to stable loci in the two genotype groups were subjected to analysis.

As can be seen from Figure 4, the genotypic effects predicted by IciMapping for the eight traits characterized by stable loci show consistently higher values in the QH type lines compared to the DW type. This observation suggests that the allelic genotype of the parental line QH-1 exerts an additive influence on these eight traits. Specifically, in the yield-related trait SL, noteworthy variations between the two types were apparent at environmental points 2023LH and 2022YL, with the QH type exhibiting superior performance by 6.55% and 9.97% (*p* < 0.01), respectively. For the NL trait, substantial differences between the two types were observed at environmental points 2023LH and 2022YL, with the QH type surpassing the DW type by 54.7% and 27.8% (*p* < 0.01), respectively. Furthermore, in the FD trait, significant differences between the two types were detected at environmental points 2023YL and 2022YL, with the QH type outperforming the DW type by 7.11% and 11.77% (*p* < 0.01), respectively. These three genotypes with notable or highly notable differences exhibit positive additive effects attributed to QH-1, corroborating the notion that the enhancing alleles at these three loci stem from the parent QH-1, as evidenced by the positive additive effect values of stable QTLs *QSL.QD-1A.1*, *QNL.QD-4B.2*, and *QFD.QD-2B* in the aforementioned investigation.

### 2.7. Candidate Gene Prediction Analysis

Firstly, gene prediction was conducted for two identified QTLs demonstrating significant enhancing effects among the nine stable loci. The stable locus *QSL.QD-1A.1* associated with the SL trait is positioned between the flanking markers *AX-109410802* and *AX-108955152* at 20.71–22.71 cM on chromosome 1A. This genomic region corresponds to 573262270 to 577688284 bp in the Chinese Spring (CS) v2.1 genome and 543283549 to 566194763 bp in the Durum Wheat (cv. Svevo) genome, where a total of 69 and 345 genes were annotated, respectively (Figure 5a, Appendix A). The stable locus *QNL.QD-4B.2* linked to the NL trait is situated between the flanking markers *AX-110031800* and *AX-108848122* at 78.59–79.09 cM on chromosome 4B. This genomic interval corresponds to 34621265 to 36338556 bp in the Chinese Spring (CS) v2.1 genome and 30286784 to 31756035 bp in the Durum Wheat (cv. Svevo) genome, with a total of 25 and 19 annotated genes, respectively (Figure 5b, Appendix A). Similarly, predictive analyses were performed for the remaining seven QTLs. In the Chinese Spring (CS) v2.1 genome, within the corresponding loci of *QPL.QD-4B.1*, *QFD.QD-2B*, *QHD.QD-2B.1*, *QHD.QD-4B*, *QKC.QD-4B.2*, *QKL-WR.QD-4B.6*, and *QKL.QD-4B.2*, 25, 110, 3, 94, 18, 15, and 18 genes were annotated, respectively. In the Durum Wheat (cv. Svevo) genome, within the corresponding loci, 19, 883, 85, 295, 14, and 295 genes were annotated, respectively (Appendix A, Appendix A).

## 3. Discussion

### 3.1. Analysis and Discussion of Enrichment of QTL Clusters in Agronomic Traits and Correlation among Agronomic Traits

In most crop localization studies, QTLs related to some agronomic traits are clustered on chromosomes, indicating that genes controlling multiple traits may be located in the same marker interval or adjacent regions. Researchers refer to this phenomenon as QTL clusters [31,32]. In addition, researchers have found a direct relationship between the enrichment of agronomic traits in QTL clusters and the traits themselves [33,34]. In this study, through the analysis of QTL positioning results in a single environment, it was found that a total of 20 QTL clusters were located on six chromosomes (Appendix A). These QTL clusters collectively contain 51 QTLs, with 7 stable QTLs distributed in 5 QTL clusters, accounting for about 77.8% of the stable QTLs detected by the ICIM-BIP method. In-depth analysis of these five QTL clusters revealed that traits such as FD and HD in cluster C5 belong to phenological traits with a significantly high correlation coefficient between them, but their correlation with the plant type trait SL in the same QTL cluster is not high. Cluster C18 contains four QTLs for FD, KW, NL, and PL, respectively. Combined with the analysis of the correlation between the corresponding agronomic traits, it was found that FD and KW are significantly negatively correlated with a correlation coefficient of −0.19; FD has no significant correlation with NL or PL; NL, with PL and KW, shows a significant positive correlation, with correlation coefficients of 0.94 and 0.34, respectively. Similarly, through the comprehensive analysis of the other three QTL clusters, it was found that there are single-locus multiple-trait effects within the QTL clusters, and the performance of agronomic traits in the field cannot be directly correlated. The magnitude of the correlation coefficient of agronomic traits and whether they are positively or negatively correlated can only be used as a reference but cannot directly determine whether the traits will appear in the same QTL cluster.

### 3.2. Exploration of the Application of Different Types of Arrays in Wheat Materials with Different Ploidies

The application of marker-assisted selection (MAS) at the whole-genome level has accelerated the breeding process of wheat. Subsequently, there have been advancements ranging from PCR-based molecular markers and extensive coverage provided by Genotyping-by-Sequencing (GBS) offered by NGS, to flexible and diverse SNP arrays. Of course, within these methods, there are certain advantages and limitations. For example, PCR-based molecular markers exhibit lower throughput and density, and many wheat traits are often influenced by multiple genes, leading to poor detection of minor genes by functional markers. Although NGS can significantly improve coverage and reduce time and cost, the poor detection ability of GBS, especially in heterozygous or early populations, is particularly evident. In addition, GBS is a time-consuming and complex process in terms of library construction, data analysis, and storage. On the other hand, SNP arrays offer great flexibility in sample and data point customizations. There are currently 9 K, 15 K, 35 K, 55 K, 90 K, 820 K, and 660 K arrays available for wheat. In the determination of SNP density distribution, the chromosomes were divided into 1 MB windows for calculation. It was observed that almost all bins in the Wheat 660 K SNP array contained SNPs, whereas the other four arrays detected many bins without SNPs (especially the Wheat 35 K, 55 K, and 90 K SNP arrays). In comparison to the Wheat 820 K SNP array, more SNPs were found at the top or end of the chromosomes in the Wheat 660 K SNP array, indicating a higher frequency of recombination events in these regions compared to the centromeric regions [35]. Thus, it is suggested that a lower coverage array such as the 55 K array could satisfy the purpose of mapping loci in a small-sized population [36]. The high-throughput SNP arrays are more suited for larger-sized fne-mapping populations or for detecting as many polymorphism loci as possible for wheat-related species in Triticeae such as Agropyron [37].

Currently, the application of the 55 K and 660 K arrays is primarily focused on hexaploid wheat, with relatively limited application in tetraploid wheat. One of the reasons for this is the reduced number and density of applied markers, which subsequently impacts the length of the genetic map and the size of the localization interval. In a study by Jiajun Liu et al., utilizing the 55 K array in common wheat (*Triticum aestivum* L.), a genetic map of an RIL population with 2524 bin markers was mapped on 34 linkage maps for the 21 chromosomes of wheat. The genetic map length was found to be 3021.04 cM, with an average of 1.20 cM per marker [28]. In a study by Ziqiang Mo, utilizing a RILs population created from tetraploid wheat materials, 150 bin markers were mapped onto 15 linkage groups across the 14 chromosomes of tetraploid wheat, with a total length of 2411.8 cM and an average marker density of 2.10 cM per bin [27]. In this study, 486 effective probes were obtained from a RIL population created from tetraploid materials using the 55 K array. The genetic map generated from these effective probes spanned 21 linkage groups, with a total genetic distance of 4397.21 cM. The average marker density between adjacent markers was 9.05 cM. Another factor contributing to the limited application on tetraploid wheat is the interference of D genome group probes during database alignment, which may lead to the inclusion of some unreliable localization intervals. These intervals often exhibit discrepancies between the information of neighboring probes, necessitating their exclusion.

### 3.3. Candidate Gene Prediction Analysis within the QTL Mapping Interval

The stable locus *QSL.QD-1A.1* on the spike length trait is located on 543283549 to 566194763 bp in the Durum Wheat (cv. Svevo) genome. Mengistu et al. identified QTL related to yield in this interval [38]. In this mapping interval, M. Graziani et al. identified QTLs regulating grain weight and test weight [39]. Within this interval, there is a potential gene, *TRITD1Av1G218480*, corresponding to Chinese Spring v2.1 genome *TraesCS1A03G1004700*, which belongs to the AP2/B3 transcription factor family protein and plays a role as an ethylene-responsive transcription factor. Wang Yuange et al. identified DUO-B1, encoding an AP2/ERF transcription factor, using wheat as a model. This transcription factor regulates the spikelet structure of bread wheat. Furthermore, mutations in DUO-B1 result in slightly more spikelets and increased grain number per spike [40]. In this interval, there is also a gene, *TraesCS1A02G413800*, which is functionally annotated as DELLA protein GAI. The function of DELLA protein in dwarfing genes is well understood, and the plant height also has a direct impact on spike length. NL is part of PL and the peduncle is part of plant height (PH); they are significantly influenced by the GA metabolism and signaling [3,7] and thus tightly linked to the PH genes. Recently, a PL QTL independent of PH was identified on 5AS [8], providing a chance to modulate the PL. In this study, *QNL.QD-4B.2* and *QPL.QD-4B.1* are located in the same interval. In this interval, we speculate that the genes *TRITD4Bv1G012860* or *TRITD4Bv1G012960* may play a regulatory role in PL and NL. These genes have a Vps4 function. Vps4 is a key component required for ESCRT-III disassembly and endosomal vesicle formation. The plant-specific ESCRT component, known as PROS, is a positive regulator of Vps4 ATPase activity. PROS interacts with Vps4 and the Vps4 positive regulator LIP5/VTA1. PROS is expressed in various tissues and cell types in Arabidopsis, and its silencing results in reduced cell proliferation and abnormal organ growth [41,42].

Synchronization of flowering time with favorable conditions for seed reproduction plays a vital role in plant species survival and is crucial for enhancing crop yield and global food security. The developmental process of wheat is intricate and governed by multiple genes categorized based on their response to vernalization (Vrn) and photoperiod (Ppd), or their influence on earliness per se (Eps). These genes orchestrate flowering time by regulating variations in the ultimate leaf count, phyllochron, and the duration of the flag leaf-anthesis phase. Notably, this study identified a previously documented locus associated with flowering time within the mapping region of *QFD.QD-2B* [43,44]. There is a corresponding region identified by previous researchers within the mapping interval of *QHD.QD-2B.1* [45]. Within the mapping interval of *QHD.QD-4B*, there is a corresponding region where previous researchers have identified a similar mapping interval [44].

KC and KL are located in the same interval (653.1 MB–654.8 MB, Durum Wheat reference genome), where known QTLs regulating grain weight are present [46]. The aspect ratio of the kernel is often directly related to the length of the kernel and the weight of the kernel. Within this range, there is a mapping interval for the aspect ratio of the kernel, as studied by Peleg and others [47,48]. In previous research reports, the interaction module between miR1432 and acyl-CoA synthetase significantly increased kernel weight by improving the grain filling rate, thereby increasing total grain yield [49]. The ZmCesA5 gene in corn encodes a glycosyltransferase protein. Overexpression of the ZmCesA5 gene results in heavier grains, while its deletion affects the metabolism of starch and sucrose, leading to a reduction in corn grain weight [50]. Therefore, we speculate that within the *QKC.QD-4B.2* and *QKL.QD-4B.2* loci, the functional annotation of *TRITD4Bv1G199600* as fatty acyl-CoA reductase may play a role in regulating KL and KC. Additionally, within the *QKL-WR.QD-4B.6* locus, the functional annotation of *TRITD4Bv1G017750* as glycosyltransferase may influence the KL-WR.

## 4. Materials and Methods

### 4.1. Plant Materials and Field Trials

In this study, two tetraploid wheat cultivars, namely “Qin Hei-1” (QH-1) and “Durum Wheat T6” (DW), were selected as the parental lines for crossbreeding. A population of 120 genotypes of recombinant inbred lines (RILs) was generated through the single seed descent method and employed as the experimental subjects.

Throughout the wheat growing seasons in Yang Ling, Shaanxi (108°07′ E, 34°30′ N) during 2020–2021 (YL21), 2021–2022 (YL22), and 2022–2023 (YL23), as well as in Lin He, Inner Mongolia (107°44′ E, 44°17′ N) during the 2021–2022 (LH22) and 2022–2023 (LH23) seasons, RILs along with the parental lines were cultivated for this study. Before sowing, the experimental fields were plowed, leveled, and fertilized. Throughout the growth phase of the plants, standard field management practices such as weeding and winter irrigation were implemented. The population was randomly segregated into three distinct small plots situated in various sections of the experimental field, with each small plot containing one replicate. Each line was sown in a single row, with a row length of 1.5 m, plant spacing set at 10 cm, and row spacing at 25 cm.

### 4.2. Phenotypic Evaluation and Statistical Analysis

Eleven significant agronomic traits related to yield were assessed across parental and population materials from diverse perspectives. These traits encompassed three plant type-related agronomic characteristics that were manually assessed post-flowering, namely, peduncle length (PL), neck length (NL), and spike length (SL). Specifically, PL denotes the distance from the spike’s base to the initial internode, NL represents the measurement from the spike’s base to the flag leaf ear, while SL indicates the length of the spike itself. Additionally, two traits linked to the growth phase, namely, flowering date (FD) and heading date (HD), were examined. For each trait, five plant materials were selected in each replicate for subsequent statistical analysis. After harvest, six grain-related agronomic traits were quantified using an SC-G type automatic grain analyzer from Hangzhou Wanshen Testing Technology Co., Ltd., China (Hangzhou, China). These traits comprised thousand-kernel weight (TKW), kernel area ratio (KAR), kernel circumference (KC), kernel length (KL), kernel width (KW), and kernel length–width ratio (KL-WR). Each line material in the population was measured five times per replicate for statistical analysis. For data processing, we referred to the method proposed by Ma et al. [22].

### 4.3. Genotyping and Genetic Map Construction

The genomic DNA (gDNA) of each RIL and the wheat 55K SNP array work were entrusted to Beijing CapitalBio Corporation (https://www.capitalbiotech.com/, accessed on 8 September 2022) for execution.

After genetic typing results were obtained, a comparison of SNP markers was conducted between the parental entities, leading to the elimination of failed markers and the identification of polymorphic markers. Additionally, SNP markers showing heterozygous genotypes and allele frequencies less than 0.3 or greater than 0.7 between the parents were further excluded. Considering that the RIL population was derived from the tetraploid wheat background and the reference was the 55K chip based on the hexaploid wheat background, the probe positions in the BIN template file were all marked as “0” in the Anchor table using the QTL IciMapping V4.1 (https://isbreeding.caas.cn/, accessed on 26 September 2023) mapping software. The redundant markers were then removed using the “By Missing” command to obtain BIN markers for constructing the genetic map. In the MAP module, a LOD value of 7.5 was selected, and the nnTwoOpt algorithm and SARF standard were used for marker sorting and rippling. The output sum file from MAP was used for mapping and labeling screening, with reference to the Chinese Spring wheat reference genome (https://urgi.versailles.inrae.fr/download/iwgsc/IWGSC_RefSeq_Assemblies/v2.1/, accessed on 12 October 2023). By comparing and filtering, erroneous values were removed or replaced, and the positioning map was ultimately limited to only the 14 chromosomes of the A and B chromosome groups. The genetic distances were converted from recombination rates to centimorgans (cM) using the Kosambi mapping function, and a genetic linkage map was drawn using MapChart V2.3.

### 4.4. Quantitative Trait Loci Mapping

QTL analysis was undertaken for the 11 agronomic traits using IciMappingV4.1 [22,23,27]. Among the detected additive QTLs, those with LOD > 3 and that explained phenotypic variation > 10% were defined as major QTLs. Additive QTLs that can be detected in two or more individual environments were also considered stable QTLs. The multi-environment trials (MET-Add) module of IcimappingV4.1 was used to identify cQTL (combined quantitative trait loci), and epistatic analysis was performed using the IciMappingV4.1 MET-EPI module and eQTL (epistatic quantitative trait loci) detection [22]. QTLs were named based on the International Rules of Genetic Nomenclature, where “QD” is the abbreviation of the parents QH-1 and DW.

For flanking sequences of key or consistent QTL data processing, we referred to the method proposed by Ma et al. [22]. Subsequently, Uniprot was employed for in-depth annotation and functional scrutiny of the candidate genes within the QTL regions (https://www.uniprot.org/, accessed on 25 October 2023) [22].

## Figures and Tables

**Figure 1 plants-13-01285-f001:**
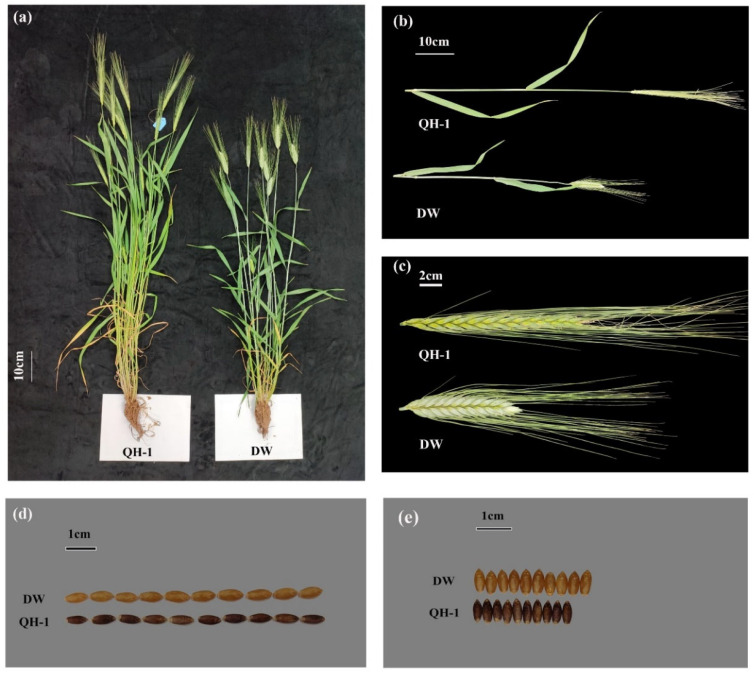
Phenotypic characteristics of the parents “QH-1” and “DW”. (**a**) Plant visual phenotype. The white line represents the scale = 10 cm. (**b**) Peduncle length (PL) and neck length (NL). Scale bar = 10 cm. (**c**) Spike length (SL). Scale bar = 2 cm. (**d**) Kernel length (KL). The black line represents the scale = 1 cm. (**e**) Kernel width (KW). Scale bar = 1 cm.

**Figure 2 plants-13-01285-f002:**
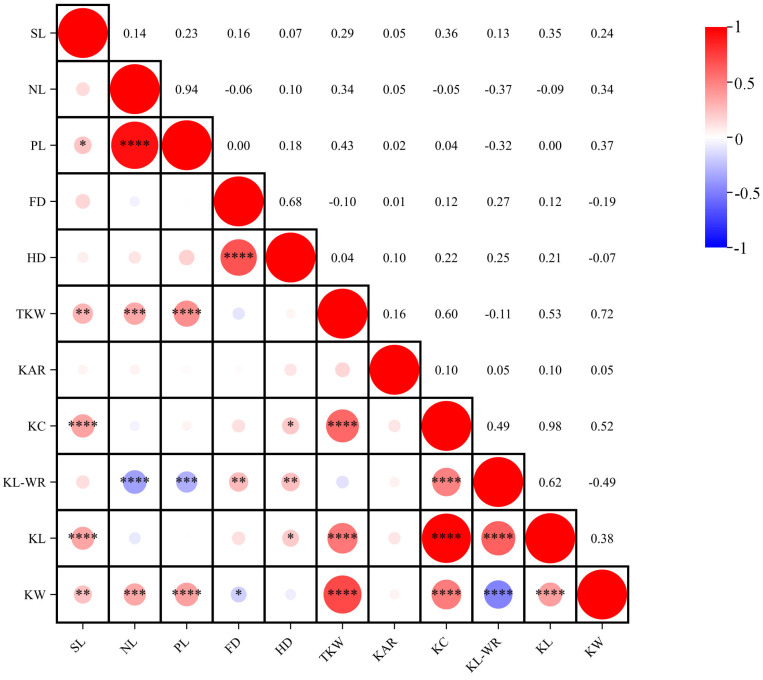
Correlation analysis of different traits based on BLUE. * Significance at the 0.05 probability level; ** Significance at the 0.01 probability level; *** Significance at the 0.001 probability level; **** Significance at the 0.0001 probability level. The colored scale on the right represents the values of correlation coefficients.

**Figure 3 plants-13-01285-f003:**
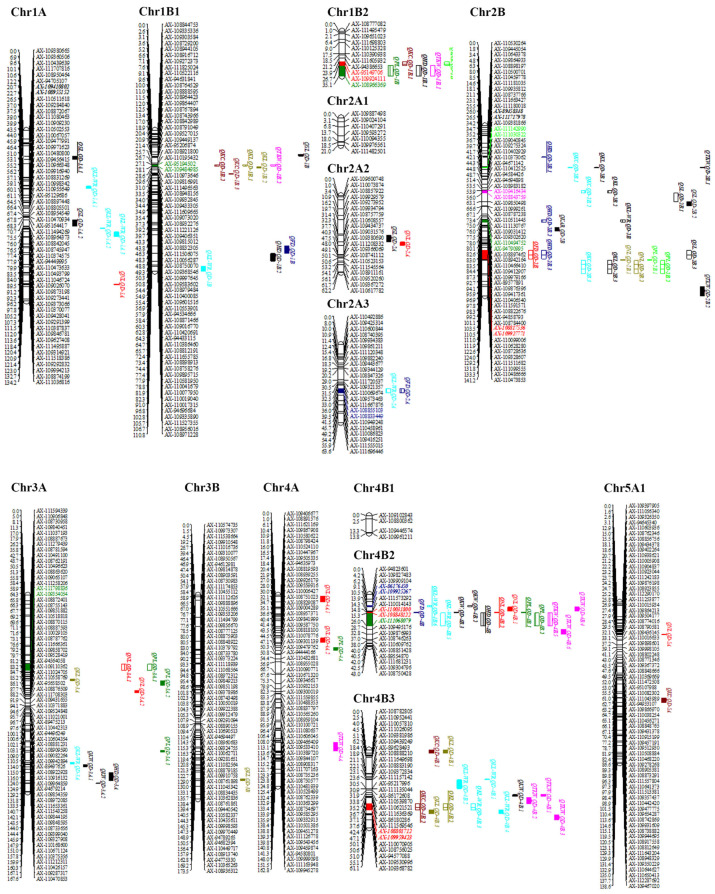
Chromosome positions of the major or stable additive QTLs for the 11 traits described. The underlined QTLs indicate that they are the stable QTLs identified in this study. Different slash types on the QTLs represent QTL clusters at different locations. The colored marker names and bars to the right of each linkage group indicate the location of the stable QTLs or QTL clusters. Different colors are used to highlight and distinguish different traits on the same chromosome.

**Figure 4 plants-13-01285-f004:**
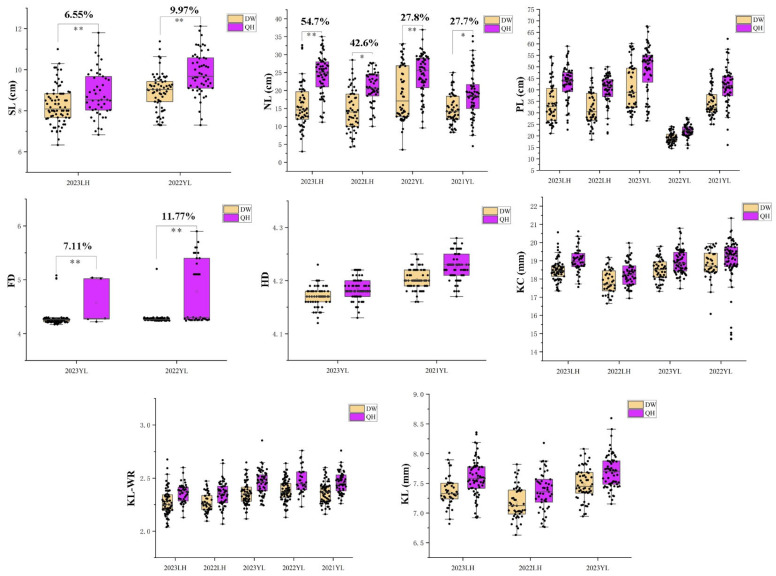
Effects of different allele genotypes of stable QTLs on correlated traits. * Significance at the 0.05 probability level; ** Significance at the 0.01 probability level.

**Figure 5 plants-13-01285-f005:**
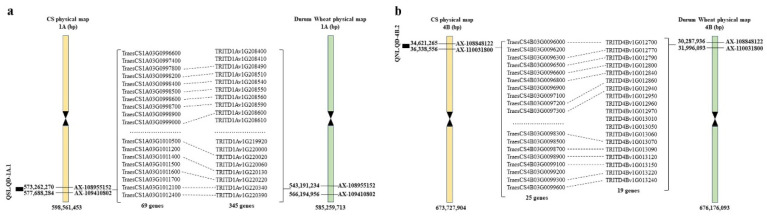
Physical maps of *QSL.QD-1A.1* (**a**) and *QNL.QD-4B.2* (**b**) on “CS” and Durum Wheat; dotted lines indicate homologous genes.

**Table 1 plants-13-01285-t001:** Phenotypic variation and heritability of characters in different environments in parents and RILs.

Traits	Year	Parental Lines		RIL Population							
		QH-1	DW	Min	Max	Mean	SD	CV%	SK	KU	*H* ^2^
SL	22LH	11.17	6.25	6.00	37.50	9.04	3.03	33.49	8.12	76.37	0.48
	22YL	12.72	9.66	7.30	12.12	9.34	1.06	11.31	0.24	−0.08	
	23LH	11.17	6.93	6.33	11.80	8.50	1.06	12.43	0.62	0.18	
	23YL	10.77	7.50	4.50	40.90	9.09	3.14	34.52	8.84	90.10	
	BLUE	11.46	7.58	6.33	19.75	9.01	1.51	16.80	3.98	24.76	
NL	21YL	21.33	16.17	4.50	35.17	17.02	5.88	34.51	0.64	0.35	0.86
	22LH	17.33	15.00	4.25	28.50	17.52	6.07	34.65	−0.25	−0.86	
	22YL	20.18	19.14	3.50	36.94	22.00	7.18	32.63	−0.27	−1.02	
	23LH	24.33	20.00	3.00	35.00	20.18	7.24	35.90	0.00	−0.87	
	23YL	31.97	16.80	9.03	46.67	24.83	11.46	46.14	3.38	24.61	
	BLUE	23.03	17.42	8.57	41.54	20.31	6.09	29.97	0.22	−0.07	
PL	21YL	46.00	35.83	16.00	62.17	38.28	8.19	21.40	0.33	0.03	0.77
	22LH	39.33	31.00	18.25	50.00	36.39	7.91	21.73	−0.37	−0.88	
	22YL	24.30	19.98	14.48	27.80	20.71	3.00	14.48	0.14	−0.47	
	23LH	45.63	37.00	21.00	59.00	38.73	9.17	23.67	−0.03	−0.97	
	23YL	55.00	35.50	24.77	67.67	45.34	10.90	24.04	−0.13	−1.17	
	BLUE	42.17	31.98	22.74	51.62	36.01	6.70	18.60	−0.12	−0.89	
FD	22YL	5.4	4.25	4.24	5.90	4.41	0.39	8.87	2.41	4.36	0.19
	23LH	6.10	6.06	5.12	6.17	6.08	0.12	1.99	−6.53	47.19	
	23YL	4.26	4.21	4.17	5.08	4.28	0.16	3.81	4.35	18.22	
	BLUE	5.54	5.13	4.83	5.85	5.21	0.16	3.16	1.93	4.01	
HD	21YL	4.20	4.19	4.16	4.28	4.21	0.03	0.61	0.35	−0.31	0.09
	22YL	4.30	4.20	4.18	5.60	4.28	0.24	5.60	4.31	17.63	
	23YL	4.19	4.15	4.12	4.23	4.18	0.02	0.53	0.02	0.14	
	BLUE	4.24	4.19	4.17	4.70	4.24	0.08	2.00	4.02	15.95	
TKW	21YL	41.82	61.09	34.50	63.89	49.42	5.53	11.18	−0.38	0.66	0.73
	22LH	46.80	50.02	31.07	71.06	47.99	4.90	10.21	1.04	5.36	
	22YL	39.06	56.20	39.92	61.64	49.74	4.34	8.73	0.01	0.00	
	23LH	48.79	52.95	38.92	59.67	48.47	4.11	8.48	0.06	0.12	
	23YL	37.70	49.40	28.12	55.46	43.90	5.13	11.68	−0.26	0.58	
	BLUE	42.83	53.93	39.20	56.88	47.90	3.27	6.83	0.22	0.34	
KAR	21YL	24.50	19.30	1.75	17.12	8.25	2.41	29.19	0.57	1.19	0.00
	22LH	6.01	10.56	1.31	18.88	10.22	3.03	29.60	0.29	0.29	
	22YL	7.89	5.52	4.02	15.12	8.40	2.71	32.34	0.44	−0.80	
	23LH	12.67	17.14	0.94	21.56	14.32	2.78	19.44	−0.85	4.49	
	23YL	10.32	9.71	7.81	21.30	13.88	2.60	18.70	0.27	−0.06	
	BLUE	12.01	12.18	6.72	13.48	10.75	1.21	11.28	−0.24	0.16	
KC	21YL	20.20	21.11	12.61	21.52	18.91	1.53	8.08	−1.45	2.86	0.48
	22LH	17.81	17.69	16.65	19.98	18.11	0.70	3.87	0.24	−0.41	
	22YL	17.73	19.70	14.69	21.34	18.89	1.21	6.40	−1.72	3.84	
	23LH	19.64	19.42	17.34	20.61	18.70	0.67	3.58	0.45	0.40	
	23YL	19.25	18.81	17.32	20.79	18.75	0.69	3.67	0.40	0.13	
	BLUE	18.89	19.31	16.95	20.56	18.64	0.63	3.36	0.08	0.23	
KL-WR	21YL	2.63	2.27	2.16	2.76	2.41	0.11	4.58	0.26	−0.09	0.82
	22LH	2.47	2.28	2.07	2.67	2.31	0.11	4.90	0.40	0.28	
	22YL	2.75	2.38	2.13	2.76	2.41	0.11	4.73	0.43	0.40	
	23LH	2.50	2.33	2.04	2.68	2.30	0.12	5.32	0.21	0.11	
	23YL	2.74	2.40	2.12	2.86	2.41	0.12	5.08	0.40	0.41	
	BLUE	2.61	2.32	2.13	2.60	2.36	0.09	3.89	0.21	0.03	
KL	21YL	7.94	8.07	5.14	8.41	7.45	0.58	7.72	−1.30	2.57	0.63
	22LH	7.28	7.09	6.63	8.18	7.27	0.31	4.32	0.31	−0.37	
	22YL	7.47	7.97	6.03	8.79	7.71	0.51	6.57	−1.36	2.81	
	23LH	8.00	7.80	6.82	8.36	7.51	0.31	4.09	0.27	0.07	
	23YL	7.97	7.67	6.94	8.60	7.60	0.30	3.96	0.35	0.39	
	BLUE	7.72	7.71	6.85	8.41	7.50	0.28	3.68	0.24	0.28	
KW	21YL	3.04	3.57	2.20	3.62	3.12	0.24	7.76	−1.08	1.67	0.49
	22LH	2.98	3.13	2.88	3.54	3.17	0.13	4.00	0.04	0.56	
	22YL	2.73	3.37	2.50	3.50	3.22	0.20	6.26	−1.69	3.01	
	23LH	3.22	3.38	2.92	3.62	3.28	0.13	3.91	−0.04	−0.04	
	23YL	2.93	3.22	2.76	3.51	3.19	0.15	4.64	−0.24	−0.11	
	BLUE	2.98	3.33	2.90	3.46	3.20	0.10	3.22	−0.06	−0.02	

*SD*: standard deviation; *CV*: coefficient of variation; *SK*: skewness; *KU*: kurtosis; *H*^2^: broad-sense heritability; *SL*: spike length (cm); *NL*: neck length (cm); *PL*: peduncle length (cm); *FD*: flowering date; *HD*: heading date; *TKW*: thousand-kernel weight (g); *KAR*: kernel area ratio (mm^2^); *KC*: kernel circumference (mm); *KL-WR*: kernel length–width ratio; *KL*: kernel length (mm); *KW*: kernel width (mm); *BLUE*: best linear unbiased estimation.

**Table 2 plants-13-01285-t002:** Stable QTLs for the quality-related trait under single-environment analysis.

Traits	QTL	Env	Interval(cM)	Left Marker	Right Marker	LOD	PVE(%)	Add
**SL**	QSL.QD-1A.1	2022YL	20.71–22.71	AX-109410802	AX-108955152	4.5567	13.5381	0.3519
		2023LH	20.71–22.71	AX-109410802	AX-108955152	9.2387	20.2021	0.4992
**NL**	QNL.QD-4B.2	2021YL	78.59–79.09	AX-110031800	AX-108848122	12.6214	34.7074	3.3629
		2022LH	78.59–79.09	AX-110031800	AX-108848122	6.6713	18.1964	2.4966
		2023LH	78.59–79.09	AX-110031800	AX-108848122	20.4944	44.2068	4.7673
		2023YL	78.59–79.09	AX-110031800	AX-108848122	70.3906	27.2688	28.1164
**PL**	QPL.QD-4B.1	2021YL	78.59–79.09	AX-110031800	AX-108848122	20.3889	52.4851	5.7827
		2022LH	78.59–79.09	AX-110031800	AX-108848122	8.3522	31.331	3.9071
		2022YL	78.59–79.09	AX-110031800	AX-108848122	20.583	43.3116	2.0306
		2023LH	78.59–79.09	AX-110031800	AX-108848122	19.2727	44.2748	6.0188
**FD**	QFD.QD-2B	2022YL	103.48–110.53	AX-108817536	AX-109927771	15.2566	29.4535	0.275
		2023YL	103.48–110.53	AX-108817536	AX-109927771	3.7689	3.8373	0.0701
**HD**	QHD.QD-2B.1	2021YL	25.99–26.47	AX-89458348	AX-111717978	4.3118	12.679	−0.0086
		2023YL	25.99–26.47	AX-89458348	AX-111717978	4.2206	14.4486	−0.0076
	QHD.QD-4B	2021YL	79.09–89.74	AX-108848122	AX-111068079	5.207	16.2552	−0.0099
		2023YL	79.09–89.74	AX-108848122	AX-111068079	3.9059	13.967	−0.0075
**KC**	QKC.QD-4B.2	2022LH	64.88–69.63	AX-108801712	AX-109959423	7.8789	21.595	−0.352
		2022YL	64.88–69.63	AX-108801712	AX-109959423	4.065	14.4427	−0.4677
		2023LH	64.88–69.63	AX-108801712	AX-109959423	7.1755	16.32	−0.2934
		2023YL	64.88–69.63	AX-108801712	AX-109959423	6.4894	16.8047	−0.3074
**KL-WR**	QKL-WR.QD-4B.6	2021YL	72.85–74.24	AX-86176450	AX-109925267	8.9608	16.7457	−0.0499
		2023YL	72.85–74.24	AX-86176450	AX-109925267	9.0497	6.2925	−0.0545
**KL**	QKL.QD-4B.2	2022LH	64.88–69.63	AX-108801712	AX-109959423	11.234	27.3212	−0.1853
		2022YL	64.88–69.63	AX-108801712	AX-109959423	4.6622	7.6794	−0.2111
		2023YL	64.88–69.63	AX-108801712	AX-109959423	5.9072	16.453	−0.1307

The underlined QTLs indicate that they are the stable QTLs and the bold font identifies that they are the major QTLs in this study. Env: Environments.

## Data Availability

The data presented in this study are available in this article and the Appendix A.

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
