# Peer review of "QTL Mapping of Yield-Related Traits in Tetraploid Wheat Based on Wheat55K SNP Array"

_plants, 2024, doi:10.3390/plants13101285_

Round 1

Reviewer 1 Report

Comments and Suggestions for Authors

The manuscript "QTL mapping of yield-related traits in tetraploid wheat based 2 on Wheat55K SNP array" presents a sound study addressing wheat crop improvement. Overall, it is scientifically sound, nicely written, results nicely presented and reference up-to-date.

A conclusion section may be added where the authors may conclude their findings and discuss the implementations of this important study. Similarly, for the convenience of readers, a few conclusive lines may also be added to Abstract, Introduction and Results sections.

The study is useful and will be of interest to the readers of Plants journal. There are no major issues, while a few minor comments are advised before acceptance.

Most of the sentences in Methods are not in the proper tense. Please  past tense.

The sentences in Lines 335-344 may be shifted to Introduction or Discussion section.

Some or parts of the tables and even figures may be shifted to supplementary materials if not too important for main text.

Comments on the Quality of English Language

Language seems correct. Just adjust the tense used in Methods section.

Author Response

Thank you very much for your guidance,Please see the attachment.

Reviewer 2 Report

Comments and Suggestions for Authors

The introduction should briefly discuss evidence on how frequently  tetraploid wheat (as studied here) and hexaploid wheat (that is better studied) share genetic variants for yield-related traits.  The literature review in the introduction does not distinguish these types.

L55 “a site independent of plant height gene correlation”  needs to be rephrased

L57 – on various chromosomes of hexploid wheat

L135 – after excluding SNPs with unsuccessful genotype calls.

L139 – I think it appropriate to report gap sizes compared to the reference. It appears to me from the supplementary table and Figure 3 that there are some fairly large gaps in the genome leading to chromosome arms that appear unlinked and some large gaps within arms.  Some of these may, of course, be caused by regions of identity between the parental lines.  Some discussion of this is important for context in interpreting and QTL results.

L142 – 26 linkage groups.  (not all groups correspond to a chromosome arm)

Table 2 – the layout of the table seems to have been corrupted during conversion to a pdf. It is a struggle to read this, but I think it is probably ok as long this does not happen in a final version. This is a case where smaller font would certainly have been preferable. I think editorial staff should have corrected this.

L162 – The colored marker names and bars to the right of each linkage group indicate the location ..

Table S4 - title in col B and others is  "Supplement Table 4 Gene and functional annotation on the reference genome of Chinese village within the localization interval."  I am pretty certain this should say "Chinese Spring"

L198 (and elsewhere) – "showcase " is not an appropriate word here. It implies an event, occasion, etc., that shows the abilities or good qualities of someone or something in an attractive or favorable way.  Better to say "demonstrate" or "show".

The Figure 4 caption seems to have become (or been combined with) the main text.  The figure caption is usually a short description of the figure and with details in the article text.

Lines 213-219.  (Hopefully) the planting environment has no effect on the genotyping but it likely does on phenotyping.  Should genotyping be replaced with phenotyping here?  Furthermore, this paragraph seems to contradict itself. Line 220 says the locus has effects on yield (I think this is based on other data not Figure 5 and an appropriate figure or table should be noted), in line 221 it says it does not have a significant impact on yield (which is not shown at all in Figure 5).  This whole paragraph needs to be rewritten or deleted.

Line 283 – note whether this QTL was in hexaploid or durum wheat

Line 286 – "In a known study" is not needed

Line 293 – they are significantly …

L 316 – The perimeter and length of the kernel are overlapping traits and .. (I am not clear what the authors mean by "range" in this section. Within the same range (min to maximum?) phenotype? Or do they mean a QTL interval. If the latter, then the paragraph should begin by stating the QTL being discussed.

L338   - I do not think it is satisfactory to identify one parent as only “Durum wheat”.  This is not a specific cultivar but a major cultivar group.  The QTLs alleles identified in this study may be specific to the cultivar.  If the origin and cultivar name of the durum wheat used is not known, then what is known should be stated such as “a cultivar identified as durum wheat XXXX in the germplasm collection of YYYYY’

L363 – clarify whether 5 “plant materials” were measured in each statistical replicate of the population or just 5 plant materials overall.  If the latter, then state how these were distributed among replicates.

L371 – were calculated across …

L372 - Correlation analysis used the https://www.chiplot.online/ website.

L382 – was hybridized with

L383 -This work was entrusted

L385-402 – this whole section should be in past tense.  You are described work that has been completed.

L396 – I don’t understand the following “The output sum file from MAP is used for library construction screening, with reference to the Chinese Spring wheat reference genome.”  Maybe this is specific to the software used. What is a library in this context?  Maybe library is the wrong word. It appears to me that they are using the output file from MAP for additional filtering to identify and exclude SNP markers not consistent with the Chinese Spring genome.

L430 – 26 linkage groups. (I think we are pretty sure that durum wheat has 14 chromosomes)

Comments on the Quality of English Language

Mostly ok - a few comments listed separately in the comments section

Author Response

(The authors gave the same response as above.)

Reviewer 3 Report

Comments and Suggestions for Authors

1). Manuscript ID: Plants_2981344

2). Manuscript Title: QTL mapping of yield-related traits in tetraploid wheat based on Wheat55K SNP array

3). Specific Comments:

The manuscript was prepared well except for minor errors. Minor editing of English language required. Overall the data was analyzed properly and figures and tables are fine and are cited in the manuscript. Please modify the manuscript based on the following comments.

 a). Materials and Methods: 

*Line 406: Change to "and calculated 1000 times"

b). Results:

*Figure 2: Please mention in the figure legend, that the colour coded scale on the right represents the correlation coefficient.

 *Figure 4: Line 211: Add this sentence at the beginning of figure legend.

 *Lines 212 to 213: Add reference supporting these sentences.

    C). Discussion:

    *Line 293: Change to "they are significantly influenced".

    *Line 295: Change to "recently"

D). Comments about ithenticate report:

 Please paraphrase the following segments of the manuscript as they are matching with the published material.

  *Lines 165 to 172; *lines 370 to 384; *lines 404 to 423

Comments on the Quality of English Language

Minor editing of English language required. Please revise the manuscript accordingly.